# Improvements in Body Composition after a Proposed Anti-Inflammatory Diet Are Modified by Employment Status in Weight-Stable Patients with Rheumatoid Arthritis, a Randomized Controlled Crossover Trial

**DOI:** 10.3390/nu14051058

**Published:** 2022-03-02

**Authors:** Erik Hulander, Helen M. Lindqvist, Anna Turesson Wadell, Inger Gjertsson, Anna Winkvist, Linnea Bärebring

**Affiliations:** 1Department of Internal Medicine and Clinical Nutrition, Institute of Medicine, Sahlgrenska Academy, University of Gothenburg, 405 30 Gothenburg, Sweden; erik.hulander@gu.se (E.H.); anna.turesson.wadell@gu.se (A.T.W.); anna.winkvist@nutrition.gu.se (A.W.); linnea.barebring@gu.se (L.B.); 2Department of Rheumatology and Inflammation Research, Institute of Medicine, Sahlgrenska Academy, University of Gothenburg, 405 30 Gothenburg, Sweden; inger.gjertsson@rheuma.gu.se

**Keywords:** rheumatoid arthritis, diet therapy, cachexia, crossover studies

## Abstract

Rheumatoid Arthritis (RA) is an autoimmune disease affecting peripheral joints. Chronic activation of inflammatory pathways results in decreased function and the development of comorbidities, such as loss of lean mass while retaining total body mass. The objective of this report was to assess whether dietary manipulation affects body composition in patients with RA as a secondary outcome. Fifty patients were included in a randomized controlled crossover trial testing a proposed anti-inflammatory Mediterranean-style diet compared to a Western diet. Body composition was measured by bioelectrical impedance spectroscopy in patients without implants (*n* = 45). Regardless of treatment, fat-free mass increased and fat mass percentage decreased during weight stability, but no differences between intervention and control in the whole group (*n* = 42, all *p* > 0.20) were found. Interaction analysis revealed that participants who were non-employed (*n* = 15) significantly decreased in fat mass (−1.767 kg; 95% CI: −3.060, −0.475, *p* = 0.012) and fat mass percentage (−1.805%; 95% CI: −3.024, −0.586, *p* = 0.008) from the intervention compared to the control period. A Mediterranean-style diet improved body composition in non-employed participants (*n* = 15). The group as a whole improved regardless of dietary allocation, indicating a potential to treat rheumatoid cachexia by dietary manipulation.

## 1. Introduction

Rheumatoid Arthritis (RA) is a chronic inflammatory disease. The primary site of symptoms is the peripheral joints, but chronic activation of inflammatory pathways as well as some anti-rheumatic pharmacological treatment (such as glucocorticoids) often result in decreased physical function and the development of comorbidities [1]. A loss of muscle mass, measured as fat-free mass (FFM), while retaining total body mass and consequently resulting in abnormally high fat mass (FM) and fat mass percentage (FM%), is common in patients with RA [2]. This state of unfavorable body composition is often referred to as rheumatoid cachexia [2]. Cachexia is a phenomenon found in a wide range of inflammatory and oncological diseases and is assumed to appear as a consequence of chronic inflammation, resulting in catabolism and loss of muscle mass [3]. In patients with RA, increased fat accumulation in tissues and the replacement of lean body mass with fat correlate to lower bone mineral density [4] as well as to worse physical function [5] and an unfavorable blood lipid profile [6].

To diagnose rheumatoid cachexia, measures of body composition such as fat-free mass index (FFMI) and fat mass index (FMI) relative to reference values have been proposed. However, there is no consensus for valid cut-off values yet. Depending on diagnostic criteria, rheumatoid cachexia is prevalent in up to a third of patients with RA [2].

Although disease-modifying anti-rheumatic drugs (DMARDs) suppress inflammation, they do not seem to reverse an established unfavorable body composition in patients with RA [7]. Resistance training has shown some success [8,9,10,11], and nutritional supplements have been tried with ambiguous results [12,13]. The efficacy of dietary treatment on rheumatoid cachexia and the effects on measures of body composition are largely unknown and thus warrant investigation.

The aim of this study was to evaluate the effects on body composition of a proposed anti-inflammatory Mediterranean-type diet, compared to a typical Western diet, in weight-stable patients with RA. We also sought to investigate if baseline characteristics modified the potential effects.

## 2. Materials and Methods

The design of the anti-inflammatory diet in a rheumatoid arthritis (ADIRA) trial has been described previously [14,15,16]. The primary outcome consisted of changes in the Disease Activity Score 28 joints Erythrocyte Sedimentation Rate (DAS28), for which results have been published [16]. The anti-inflammatory effect of the anti-inflammatory diet has been demonstrated, and the effect on biomarkers of inflammation is published elsewhere (15). This paper examines changes in body composition as a secondary outcome.

### 2.1. Ethical Statement

The ADIRA trial was approved by the regional ethical review board in Gothenburg (976-16 and T519-17), registered on ClinicalTrials.gov (NCT02941055), and all participants provided signed informed consent prior to enrolment. All procedures were consistent with the Helsinki Declaration.

### 2.2. Recruitment

Outpatients with manifest RA according to 1987 American College of Rheumatology and 2010 American College of Rheumatology/European Alliance of Associations for Rheumatology criteria [17] at Sahlgrenska University Hospital, Gothenburg, Sweden, were identified through the Swedish Rheumatology Quality Register. Those who resided in areas where home delivery of foods was possible were invited by post.

Inclusion criteria were a DAS28 of at least 2.6, unchanged DMARD medication for the past eight weeks prior to screening, 18–75 years of age, and a minimum of two years disease duration. Patients were excluded if they had life-threatening diseases or were pregnant or lactating, allergic to any of the trial foods, or unable to understand the study information.

### 2.3. Study Design

To reduce between-subject variation in this heterogeneous population, a crossover design was chosen. A computer-generated list was used to randomize participants to begin with either intervention or control diet (allocation ratio 1:1). Participants received food items with recipes weekly throughout both diet periods and dietary advice at the beginning of each diet period. The diet periods lasted for 10 weeks with a 4-month washout. The study ran in two batches, where individuals in the first group began entering the trial in February 2017 and individuals in the second group in August 2017.

### 2.4. Dietary Interventions

Participants received food bags designed to provide 1100 kcal daily for 5 days per week, to replace approximately half of their usual daily dietary intake. During both diet periods, daily breakfasts, snacks, and one main meal were provided. Study staff encouraged participants to keep weight-stable throughout the study.

During the intervention diet period a proposed Anti-inflammatory Mediterranean-type diet was provided including a breakfast containing a probiotic fruit drink (2 × 10^10^ colony forming units of Lactobacillus plantarum 299 v), frozen berries and either low fat sour milk with walnuts and whole grain muesli, fiber-enriched oat porridge and walnuts, or low-fat yogurt and whole-grain muesli. The whole-grain muesli consisted of 51% seeds and nuts. As a main meal, fish was provided 3–4 times weekly, and 1–2 meals were vegetarian. Vegetables and either whole grains or potatoes were provided with every meal. As daily snacks, two fruits were supplied. Canola oil was provided for cooking. For meals not provided, dietary advice was given to limit red meat to ≤3 times per week; keep fruit, berries, and vegetable intake to ≥5 portions per day; and choose whole-grain products, low-fat dairy products, margarine, and vegetable oils.

The control diet was designed to reflect average nutritional intake in Sweden—i.e., a typical western diet. Breakfast consisted of orange juice and either a mix of quark and yogurt served with corn flakes or white bread with butter and cheese. The main meals included red meat 3–4 times weekly and chicken 1–2 times weekly. The meals contained a comparatively smaller portion of vegetables and refined grains or potatoes. As a daily snack, a protein bar or a serving of quark curd was supplied. For meals not provided, participants were advised to consume red meat ≥ 5 times per week; fish ≤ 1 time per week; and fruits, berries, and vegetables ≤ 5 portions daily; and to choose whole-fat dairy products and butter and avoid probiotic products.

To blind participants, study staff referred to the dietary regimens as two experimental nutritional therapies labeled either “fiber diet” (intervention) or “protein diet” (control) in communication with participants.

### 2.5. Data Collection

Questionnaires assessing lifestyle, physical activity, habitual dietary intake, and demographical data were filled out by participants prior to entering the trial. Based on data from food frequency questionnaires, a dietary quality index as previously described by the Swedish National Food Agency was calculated [18]. Height, waist, and hip circumference was measured to the closest 0.5 cm, and weight was measured in light clothing, with 1 kg subtracted to account for remaining clothes.

Before and after each diet period, body composition was measured in participants without metallic surgical implants in the fasting state by bioimpedance spectroscopy using ImpediMed SFB7 (ImpediMed, Brisbane, Australia). Measurements were performed according to manufacturer’s instructions on the participant’s right side (foot and hand). All participants were in a supine position for 5 min prior to measurement. FM and FFM were calculated in the manufacturer’s software BioImp Body Composition Analysis Software Version 5.5.0.1. In order to categorize participants as having an unfavorable body composition, cut-off values in FMI and FFMI for rheumatoid cachexia as proposed by Engvall et al. [19] and Elkan et al. [20] were used. Additionally, a high FMI was defined as FMI above the 75th percentile, and low FFMI as FFMI below the 25th percentile. Classifications of body composition were in all cases compared to a Swiss reference population [21].

Three-day food records were collected and analyzed by a registered dietitian. From data on nutritional intake, Nutrient Rich Foods index (NRF) 11.3 was calculated as previously described elsewhere [22]. Further, experienced nurses assessed DAS28 as a marker of disease activity, and participants filled out a Swedish version of the Health Assessment Questionnaire (HAQ) [23]. Compliance was assessed by mid-period interviews by telephone, where a score was calculated based on how many of the intervention/control meals had been consumed in full (scored 100%), in part (scored 50%), or not at all (scored 0%) during the past week. Participants with a mean compliance score above 80% were considered as compliant. Questionnaires on medication use were filled out during screening, and changes were recorded by participants during each dietary period.

Fasting blood samples were taken by venipuncture and serum and plasma were separated. All samples were stored in −80 °C until analysis.

### 2.6. Laboratory Analyses

Concentrations of CRP and ESR were measured by routine analysis in fresh samples at Sahlgrenska University Hospital (Gothenburg, Sweden). Albumin and insulin-like growth factor-1 (IGF-1) were quantified in serum following routine procedures by the clinical laboratory at Sahlgrenska University Hospital.

Amino acids in serum were quantified by ^1^H Nuclear Magnetic Resonance (NMR) analysis. Samples were prepared according to In Vitro Diagnostics Research (IVDr) standard operating procedures as described by Dona et al. [24].

In brief, serum samples were thawed at room temperature for 30 min, then centrifuged at 3500× *g* for 1 min at 4 °C. Thereafter, 325 μL of serum was transferred with a SamplePro L liquid handler (Bruker BioSpin) to a deepwell plate (Porvair, cat. no 53.219030) containing 325 μL NMR buffer ((75 mM sodium phosphate, pH 7.4, 0.08% 3-(trimethylsilyl) propionic-2,2,3,3-d_4_), 0.04% sodium azide, 20% *v*/*v* D_2_O) per well. The plate was shaken at 400 rotations per minute, 12 °C for 5 min in a Thermomixer Comfort (Eppendorf). Finally, 600 μL sample was transferred to 5 mm SampleJet NMR tubes with the SamplePro L. The sample tubes, deepwell plate, and SampleJet rack were kept at 2 °C during the preparation in the SamplePro L robot. ^1^H NMR data was acquired on a Bruker 600 MHz Avance III HD spectrometer equipped with a room temperature 5 mm BBI probe and a cooled SampleJet sample changer. In brief, 1D NOESY (‘noesygppr1d’ pulse sequence), 1D CPMG (‘cpmgpr1d’), and 2D J-resolved (‘jresgpprqf’) spectra were acquired according to the standard IVDr parameter settings at 310 K. Absolute quantification of amino acids in serum was provided by automated deconvolution of the 1D NOESY data sets through the use of the B.I. Quant-PS 2.0.0 service (Bruker BioSpin). Experimental parameters are available upon request.

### 2.7. Statistical Analysis

For comparison between baseline values, since not all numerical variables showed a normally distributed pattern, the Mann-Whitney test was used. For comparison of proportions at baseline, Fisher’s Exact Test was used. Analyses of treatment effects on body composition were performed by a linear mixed ANCOVA model. The fixed variables were dietary treatment (intervention or control), period (the first or second diet period), sex, and the baseline value for each outcome variable. Individual participant ID was included as a random effect. Residuals were inspected and conformed to model assumptions without the need for variable transformation.

Potential interaction between pre-defined baseline variables and diet in their effects on the outcomes FM, FM%, and FFM were explored to identify effect modification by adding interaction terms to the linear mixed ANCOVA model. Variables tested for effect modification were dichotomized values of baseline dietary quality (measured as an index ranging from 0–12), physical activity (an index with range 1–4 was calculated from questions on weekly intentional physical exercise and activity during daily life), high FMI, low FFMI, low FFMI and high FMI combined, HAQ, age, educational level (low defined as ≤2-year senior high school, high as >2-year senior high school or college/university education), albumin concentration, and employment status (working some hours per week/not working at all).

All tests were performed using SPSS Statistics version 25 (IBM, Armonk, NY, USA). In all statistical tests, significance level α was set at 0.05, except for interaction analyses where significance was accepted at 0.20.

### 2.8. Power Calculation

The power calculation was based on expected changes in the primary outcome of the trial, DAS28. In order to detect a change of 0.6 units in DAS28 with 90% power and α = 0.05, a sample size of 38 patients was needed, and to account for dropouts 50 patients were recruited.

## 3. Results

### 3.1. Subjects

Of the 50 participants entering the trial, composition data were available for 45; of those, 42 completed at least one diet period and 40 completed the entire study (Figure 1). The majority of participants were women, with a median age 62 years, of European descent and a high educational level (Table 1). At baseline, 31% of participants were overweight and 26% were obese. A minority of participants, 36%, had a healthy waist-to-hip ratio (defined as <0.8 for females and <0.9 for males). Rheumatoid Cachexia was prevalent in 17–29% of participants depending on diagnostic criteria (as defined by Engvall et al. [19] and Elkan et al. [20]) (Table 1).

Among the participants completing at least one diet period, there were 17 reports (21% of the diet periods) of gastrointestinal discomfort, of which 13 occurred during the intervention period.

### 3.2. Effects of Diet on Body Composition

In analysis of effects of diet on body composition in the whole group (*n* = 42), there were no differences between intervention and control diet period in any measure of body composition (Appendix A). FFM and FFMI increased and FM% decreased during both the control and the intervention periods, whereas FM and FMI decreased during the intervention period only. Weight did not change significantly between or during any of the diet periods.

Interaction analysis revealed that changes in FM and FM% between the intervention and control periods were moderated by baseline employment status. Participants not employed (i.e., not working at all, *n* = 15) experienced decreases in FM, FMI, and FM% during the intervention compared to the control period (Table 2). In contrast, no changes in body composition between intervention and control periods were seen among employed participants. Among biomarkers hypothesized to relate to body composition, BCAA concentration was higher after the control period compared to after the intervention period among employed participants only (Table 2). Concentration of IGF-1 was not affected either within or between diet periods (Table 2).

At baseline, those not employed were older and had higher FMI and FM%, higher nutrient density, and higher intake of marine omega-3 fatty acids. In addition, among non-employed participants a higher proportion were female and a lower proportion used csDMARDs (Appendix A).

The results were consistent when analyzing only participants with high compliance to both diet periods who also did not discontinue or begin on a new DMARD or glucocorticoid treatment (all *n* = 27, employed *n* = 16, not employed *n* = 11) (data not shown).

## 4. Discussion

This investigation examined effects on body composition from a proposed anti-inflammatory, isocaloric Mediterranean-style diet regimen compared to a control diet resembling the average nutritional intake in Sweden in a randomized controlled crossover trial on patients with manifest RA.

We found no significant effects between the intervention and the control period in measures of body composition in the group as a whole. However, baseline employment was identified as an effect modifier, where non-employed participants showed improvement in body composition after the intervention diet period compared to after the control diet period. This was not the case among employed participants.

We had hypothesized that participants who did not regularly work would have a less structured food intake and therefore reap greater benefits from a health-promoting diet intervention with home delivery of foods. At the study baseline, intake of marine omega-3 fatty acids was higher among non-employed participants, indicating a higher marine food intake. Participants without employment also differed from the rest in terms of age, nutrient density, medication, and gender and body composition.

Although many factors could differ between individuals who work and those who do not, none of the other examined baseline variables were indicated as effect modifiers. Furthermore, none of the metabolic markers that we included in our analysis indicated significant changes within the group of non-employed participants. The reasons for the observed interaction between dietary treatment and employment status are thus unknown and could be multifactorial. It is also possible that responders to the diet intervention clustered by chance in the non-employed subgroup.

In the group as a whole, without stratification by employment status, there were significant improvements in body composition over time regardless of treatment. Our data do not allow us to draw conclusions on the causality of these improvements. It is possible that overall nutritional intake was improved by home-delivered, easy-to-prepare, healthful menus regardless of diet allocation. It is also possible that participation in an RCT increased awareness and provoked health-promoting behaviors beyond the scope of the planned nutritional treatment.

Earlier dietary intervention studies on body composition in patients with RA are rather scarce. Marcora et al. compared two different amino acid supplements in a randomized placebo-controlled double-blinded trial but saw no difference between treatments [12]. The authors found favorable effects over time and hypothesized that an increased nitrogen intake as a result from both interventions could explain the effect. However, a weakness when interpreting improvements over time disregarding treatment allocation is the inability to control for attention–placebo effects.

The ADIRA trial has several strengths; we aimed at controlling for weight change by supplying an isocaloric diet intervention and gave instructions to participants to try keep their weight stable. The crossover design limits inter-individual variation and likely gives higher statistical power. Furthermore, participants had a median washout of 4 months, which we expect to suffice for a return to approximately habitual dietary intake at the beginning of the second diet period. We measured body composition following a standardized protocol in the fasting state after 5 min in a supine position. Moreover, our trial ran in two batches and lasted over a whole year; seasonal variations are thus not likely to explain the effects. There are also limitations that need to be addressed. Firstly, we did not collect information on physical activity throughout the trial (only at baseline); consequently, we cannot adjust for potential changes in physical activity. Furthermore, our statistical power calculation was not designed to detect changes in body composition; it is possible that the results would be clearer had we recruited more participants. Lastly, body composition was assessed through bioimpedance by multiple operators; the precision of the measurements would probably have improved if a single operator or a more robust method such as Dual Energy X-ray Absorptiometry had been used.

## 5. Conclusions

In conclusion, employment status modified the effect of a Mediterranean-style diet intervention with home-delivered, easy-to-prepare food bags compared to a control diet resembling a Western diet. The intervention diet period significantly improved body composition in non-employed participants, while no effect was seen among those with employment. Our results indicate a potential to alleviate rheumatoid cachexia by dietary intervention and motivate further studies exploring mechanisms, causation, and long-term effects.

## Figures and Tables

**Figure 1 nutrients-14-01058-f001:**
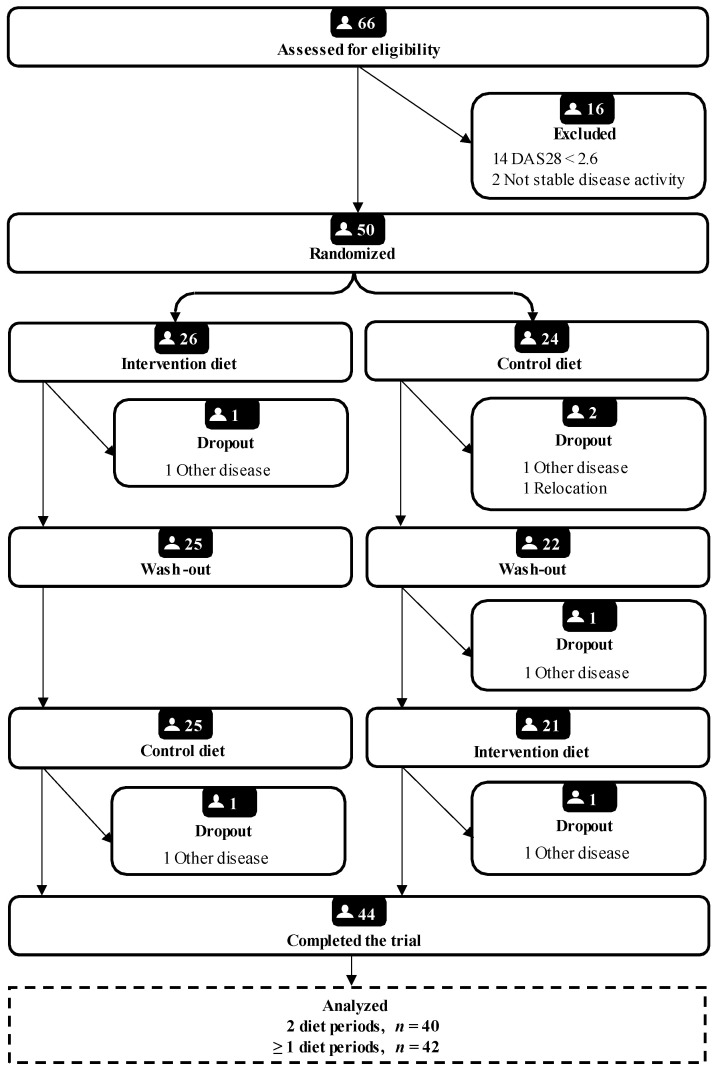
Flow chart of subject recruitment in the ADIRA trial reported according to CONSORT DAS28, Disease Activity Score 28 joints Erythrocyte Sedimentation Rate.

**Table 1 nutrients-14-01058-t001:** Baseline data of participants who completed at least one diet period, stratified by sequence according to the CONSORT guidelines.

	All Participants *n* = 42	Intervention-Control *n* = 22	Control-Intervention *n* = 20	*p*
	Median (IQR)	Median (IQR)	Median (IQR)	
Age (year)	62.3 (50.6, 70.2)	62.3 (56.7, 68.4)	62.2 (46.5, 72.2)	0.852 ^a^
Disease duration (year)	18.3 (10.3, 24.6)	17.7 (9.6, 28.3)	19.0 (10.8, 23.2)	0.808 ^a^
Weight (kg)	76.6 (66.5, 83.3)	72.7 (65.4, 85.2)	78.0 (69.5, 82.7)	0.649 ^a^
BMI (kg/m^2^)	25.3 (23.6, 30.2)	25.9 (22.7, 30.5)	25.3 (23.8, 30.2)	0.911 ^a^
FFMI (kg/m^2^)	17.5 (15.3, 18.6)	16.7 (15.1, 18.6)	17.8 (15.3, 18.7)	0.609 ^a^
FMI (kg/m^2^)	9.5 (6.6, 12)	9.9 (6.7, 12.0)	9.2 (6.5, 11.0)	0.663 ^a^
FM (%)	35.1 (29.1, 42.0)	36.1 (28.7, 43.2)	35.1 (30.7, 39.9)	0.493 ^a^
Waist-Hip ratio	0.84 (0.79, 0.9)	0.83 (0.79, 0.89)	0.85 (0.81, 0.91)	0.627 ^a^
DAS28	3.62 (3.02, 4.65)	3.70 (3.08, 4.57)	3.46 (2.91, 4.66)	0.794 ^a^
HAQ	0.50 (0.13, 1.25)	0.50 (0.13, 0.94)	0.69 (0.13, 1.47)	0.294 ^a^
Albumin (g/L)	39 (37, 41)	38 (37, 40)	39 (36, 42)	0.455 ^a^
IGF-1 (µg/L)	112 (87, 143)	106 (84, 139)	118 (95, 157)	0.326 ^a^
Dietary intake				
Dietary Quality Index	7.0 (5.0, 7.3)	7.0 (5.8, 8.0)	5.5 (5.0, 7.0)	0.082 ^a^
Energy (kcal)	1730 (1390, 2150)	1730 (1340, 2070)	1790 (1420, 2220)	0.663 ^a^
Protein (E%)	15.4 (13.9, 17.4)	15.3 (14.1, 18.7)	15.5 (13.7, 16.9)	0.990 ^a^
Carbohydrate (E%) (including fiber)	42.5 (38.3, 49.2)	43.7 (40.0, 51.8)	41.7 (37.0, 47.8)	0.149 ^a^
Fat (E%)	36.1 (31.7, 40.1)	35.2 (31, 38.1)	36.8 (35.4, 41.4)	0.116 ^a^
Protein (g/kg bodyweight/day)	0.89 (0.72, 1.14)	0.89 (0.80, 1.14)	0.91 (0.67, 1.32)	0.990 ^a^
Protein (g/day)	67.1 (56.5, 78.6)	68.2 (56.5, 77.1)	66.5 (52.9, 81.1)	0.847 ^a^
Fiber (g/day)	18.0 (13.7, 21.1)	17.5 (13.8, 22.0)	19.8 (13.7, 20.9)	0.799 ^a^
Meal frequency (meals per day > 25 kcal)	5.0 (4.0, 5.3)	5.0 (4.0, 6.0)	4.8 (3.8, 5.3)	0.891 ^a^
	*n* (%)	*n* (%)	*n* (%)	
Medication				
bDMARD usage	17 (41)	8 (36)	9 (45)	0.754 ^b^
csDMARD usage	32 (76)	18 (82)	14 (70)	0.477 ^b^
Measures of rheumatoid cachexia				
Engvall ^2^	7 (17)	4 (18)	3 (15)	1.0000 ^b^
Elkan ^3^	12 (29)	6 (27)	6 (30)	1.0000 ^b^
High FMI ^4^	23 (55)	13 (59)	10 (50)	0.7569 ^b^
Low FFMI ^5^	15 (36)	8 (36)	7 (35)	1.0000 ^b^
Low FFMI & high FMI ^6^	9 (21)	5 (23)	4 (20)	1.0000 ^b^
Sex				
Female	34 (81)	18 (82)	16 (80)	1.000 ^b^
Parental origin				1.000 ^b^
Europe	39 (93)	20 (91)	19 (95)	
Africa	1 (2)	1 (5)	0 (0)	
Asia	2 (5)	1 (5)	1 (5)	
Smokers	2 (5)	2 (9)	0 (0)	
Employment status				0.571 ^b^
Not employed	15 (36)	6 (27)	9 (45)	
Employed < 15 h/week	2 (5)	2 (9)	0 (0)	
16–30 h/week	6 (14)	3 (14)	3 (15)	
31–40 h/week	8 (19)	4 (18)	4 (20)	
>40 h/week	11 (26)	7 (32)	4 (20)	
Educational level				0.611 ^b^
Junior high school	5 (12)	2 (9)	3 (15)	
2 year senior high school	8 (19)	4 (18)	4 (20)	
≥3 year senior high school	6 (14)	2 (9)	4 (20)	
College or university	23 (55)	14 (64)	9 (45)	
Physical activity during everyday life				0.052 ^b^
Light	5 (12)	4 (18)	1 (5)	
Light but partly active	14 (33)	10 (46)	4 (20)	
Light and active	13 (31)	6 (27)	7 (35)	
Sometimes physically heavy	10 (24)	2 (9)	8 (40)	
Physically heavy most of the time	0 (0)	0 (0)	0 (0)	
Intentional physical exercise				0.828 ^b^
Never	6 (14)	2 (9)	4 (20)	
Now and then, not regularly	13 (31)	7 (32)	6 (30)	
1–2 times/week	9 (21)	6 (27)	3 (15)	
2–3 times/week	8 (19)	4 (18)	4 (20)	
>3 times/week	6 (14)	3 (14)	3 (15)	

bDMARD, biological disease modifying anti-rheumatic drug; BMI, Body mass index; csDMARD, conventional synthetic disease modifying anti-rheumatic drug; DAS28, Disease Activity Score 28 joints Erythrocyte Sedimentation Rate; FFMI, Fat free mass index; FM, Fat mass; FMI, Fat mass index; HAQ, Health assessment questionnaire disability index; IGF-1, Insulin-like growth factor 1; NRF11.3, Nutrient Rich Foods index 11.3. Docosahexaenoic, docosapentaenoic and eicosapentaenoic acid combined. ^2^ Defined as FFMI < 10th percentile and FMI > 25th percentile as proposed by Engvall et al. [19]. ^3^ Defined as FFMI < 25th percentile and FMI > 50th percentile as proposed by Elkan et al. [20]. ^4^ Defined as FMI > 75th percentile. ^5^ Defined as FFMI < 25th percentile. ^6^ Defined as FFMI < 25th and FMI > 75th percentile. Reference for body composition ranges is taken from Schutz et al. [21]. ^a^ Mann-Whitney Test using the exact *p*-value. ^b^ Fisher’s exact test.

**Table 2 nutrients-14-01058-t002:** Modelled estimates of differences in body composition, and metabolic markers hypothetically related to body composition, between intervention and control diet periods among patients with RA in the ADIRA trial stratified by employment status ^1^.

	Not Employed (*n* = 15)				Employed (*n* = 27)			
	Intervention	Control	Difference between Periods		Intervention	Control	Difference between Periods	
	Mean (95% CI)	Mean (95% CI)	Mean (95% CI)	*p* ^2^	Mean (95% CI)	Mean (95% CI)	Mean (95% CI)	*p* ^2^
Measures of body composition								
Weight (kg)	−0.436 (−1.191, 0.320)	0.376 (−0.413, 1.165)	−0.812 (−1.961, 0.337)	0.149	−0.211 (−1.006, 0.584)	0.027 (−0.749, 0.803)	−0.238 (−1.362, 0.885)	0.672
BMI (kg/m^2^)	−0.168 (−0.451, 0.116)	0.143 (−0.153, 0.440)	−0.311 (−0.736, 0.115)	0.137	−0.051 (−0.328, 0.226)	−0.02 (−0.290, 0.251)	−0.031 (−0.423, 0.360)	0.873
FFM (kg)	0.876 (0.016, 1.735)	0.240 (−0.644, 1.125)	0.636 (−0.259, 1.531)	0.149	0.960 (0.006, 1.913)	1.046 (0.114, 1.978)	−0.087 (−1.369, 1.196)	0.890
FFMI (kg/m^2^)	0.317 (−0.007, 0.641)	0.090 (−0.243, 0.423)	0.227 (−0.108, 0.561)	0.167	0.324 (0.011, 0.638)	0.34 (0.033, 0.646)	−0.015 (−0.437, 0.406)	0.940
FM (kg)	−1.312 (−2.151, −0.472)	0.456 (−0.422, 1.333)	−1.767 (−3.060, −0.475)	0.012	−0.829 (−2.183, 0.524)	−1.403 (−2.725, −0.081)	0.573 (−1.346, 2.492)	0.551
FMI (kg/m^2^)	−0.488 (−0.807, −0.170)	0.172 (−0.161, 0.505)	−0.660 (−1.140, −0.180)	0.011	−0.267 (−0.705, 0.171)	−0.465 (−0.893, −0.038)	0.198 (−0.422, 0.819)	0.523
FM (%)	−1.653 (−2.672, −0.634)	0.152 (−0.905, 1.209)	−1.805 (−3.024, −0.586)	0.008	−1.231 (−2.532, 0.071)	−1.922 (−3.193, −0.651)	0.691 (−1.159, 2.541)	0.456
Biomarkers								
BCAA ^3^ (mmol/L)	−0.012 (−0.044, 0.019)	0.003 (−0.029, 0.036)	−0.015 (−0.058, 0.027)	0.426	−0.022 (−0.043, −0.001)	0.008 (−0.013, 0.029)	−0.030 (−0.054, −0.005)	0.022
IGF-1 (µg/L)	1.379 (−7.255, 10.013)	1.233 (−7.794, 10.260)	0.146 (−12.607, 12.900)	0.981	−0.894 (−9.854, 8.067)	5.077 (−3.676, 13.830)	−5.970 (−18.666, 6.725)	0.349

BCAA, Branched chain amino acids; BMI, Body mass index; FFM, Fat free mass; FFMI, Fat free mass index; FM, Fat mass; FMI, Fat mass index; IGF-1, Insulin-like growth factor 1. ^1^ Participants completing at least one diet period and where measurement of body composition was possible (*n* = 42). RA, Rheumatoid Arthritis; ADIRA, Anti-inflammatory Diet In Rheumatoid Arthritis. Analysed by use of a linear mixed model with period, treatment, sex, and baseline value as fixed effects and subject as random effect. ^2^ Intervention–Control values. ^3^ Leucine, isoleucine, and valine combined.

## Data Availability

Not applicable.

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
