# Peer review of "Improvements in Body Composition after a Proposed Anti-Inflammatory Diet Are Modified by Employment Status in Weight-Stable Patients with Rheumatoid Arthritis, a Randomized Controlled Crossover Trial"

_nutrients, 2022, doi:10.3390/nu14051058_

Round 1

Reviewer 1 Report

In this study, Erik Hulander et al. assessed whether dietary manipulation affects body composition in patients with RA as a secondary outcome. While it's an interesting topic, the study was not well designed and data wasn't clearly presented. 

Major comments

  1. The author studies a proposed anti-inflammatory diet in rheumatoid arthritis patients. However, there was no measurement of inflammation in any means in the trial. The title and how the study actually be conducted may be misleading. The author should determine if the diet used in the study is anti-inflammatory!
  2. How the sample size was determined? 
  3. The compliance of participants was not carefully discussed or considered in statistical analysis.

Author Response

In this study, Erik Hulander et al. assessed whether dietary manipulation affects body composition in patients with RA as a secondary outcome. While it's an interesting topic, the study was not well designed and data wasn't clearly presented. 

To the best of our knowledge, this is one of very few studies examining the effect of dietary intervention on body composition in patients with RA. We agree that it would be beneficial to study the effects over a longer period of time, still, we believe our results are highly relevant to the medical community. We regret that the reviewer considers the study not well designed, but since this comment isn’t elaborated on, it is difficult to know what the reviewer refers to. A cross-over design is usually regarded as a good study design for dietary interventions, but can maybe be discussed for the presented outcome (body composition). However, within this group of patients a parallel design would have demanded a very large patient group since these patients differ not only in lifestyle but also in disease progression, comorbidities and medical treatment. In fact, the quality of the dietary intervention study is high, as foods were provided to participants and compliance was high.  

Our data indicate that body composition in patients with RA improved by a dietary intervention. This must of course be further investigated in larger and longer studies, but it still implies that the state of rheumatoid cachexia does not necessarily have to be part of the natural disease progression and can be reversed through lifestyle changes.

Major comments

1.The author studies a proposed anti-inflammatory diet in rheumatoid arthritis patients. However, there was no measurement of inflammation in any means in the trial. The title and how the study actually be conducted may be misleading. The author should determine if the diet used in the study is anti-inflammatory!

We apologize for not stating this clearer. We have now clarified that the anti-inflammatory effect has been demonstrated and data on effect on biomarkers of inflammation has published, please see line 60-61.

2.How the sample size was determined? 

Please see line number 192-195.

3.The compliance of participants was not carefully discussed or considered in statistical analysis.

We have added a section where we present how we evaluated compliance, please see line 136-141, and we have added a sentence in results, please see line 251-254.

Reviewer 2 Report

The study is well designed and the results are interpreted clearly.

I have one comment about the types of diet. The authors have mentioned the "Mediterranean diet" and "Western diet" in the study. The authors need to explain more about these types of diets.

Author Response

I have one comment about the types of diet. The authors have mentioned the "Mediterranean diet" and "Western diet" in the study. The authors need to explain more about these types of diets.

Thank you for this comment. We were not clear enough in our presentation of the diet. Please see line 87-111.

Round 2

Reviewer 1 Report

The authors have addressed my concerns in the revision.